# Selected Biotopes of *Juniperus communis* L. in Slovakia and Their Chemotype Determination

**Ivan Salamon** [1],*, **Pavol Otepka** [2], **Maryna Kryvtsova** [3], **Oleh Kolesnyk** [4] and **Myroslava Hrytsyna** [5]

1. Department of Ecology, Faculty of Humanities and Natural Science, University of Presov, 17th November St. 01, SK-081 16 Presov, Slovakia
2. European Ecocycles Society, Matrai St. 36, 3200 Gyongyos, Hungary; pavol.otepka@gmail.com
3. Department of Genetic, Plant Physiology and Microbiology, Faculty of Biology, Uzhhorod National University, Voloshina, 32, 88000 Uzhhorod, Ukraine; maryna.krivcova@uzhnu.edu.ua
4. Department of Botany, Faculty of Biology, Uzhhorod National University, Voloshina, 32, 88000 Uzhhorod, Ukraine; oleg.kolesnyk@uzhnu.edu.ua
5. Department Pharmacy and Botany, Faculty of Social Development and Health, Stepan Gzhytskyj National University of Veterinary Medicine and Biotechnologies, Pekarska Str., 50, 79010 Lviv, Ukraine; hrytsynamr@lvet.edu.ua
* Correspondence: ivan.salamon@unipo.sk; Tel.: +421-917984060

**Abstract:** The objective of this work was to map population of common juniper (*Juniperus communis* L.) in the territory of Slovak Republic. Common juniper is not protected by Slovakian law or the authorities; therefore, there is no law to preserve these plant populations. Biotopes of common juniper consist of light- to heavy-density trees and shrubs. The expansion of its habitat is also connected to human involvement. The loss of juniper naturally in observed plants is caused by a change in land use, loss of feeding pastures for cattle and sheep, and eutrophication of the environment. The current study was focused on monitoring the population of this plant species in the years of 2018–2020, the isolation of essential oils, and the identification of qualitative and quantitative characteristics. It was confirmed that juniper berries usually contain from $0.5 \pm 0.05$ to $1.8 \pm 0.06\%$, usually $1.2 \pm 0.16\%$, volatile oil depending on geography, altitude, ripeness, and other factors. Volatile oil is made up mostly of monoterpenes, mainly α-pinene (from $37.60 \pm 2.23$ to $61.00 \pm 0.60\%$), β-myrcene (from $8.03 \pm 2.02$ to $10.56 \pm 0.05\%$), and sabinene (from $3.50 \pm 0.30$ to $22.0 \pm 0.96\%$). The dendrogram was constructed after a hierarchical cluster analysis based on the essential oil substances, which showed four different confirmed chemotypes. The essential oil is widely used in medicines, perfumes, insect repellents, insecticides, shoe polish, and in microscopy as a clearing agent of an immersion oil. The quality and chemotypes of juniper berries are very important for the Slovak national beverage "Borovicka" and the distillery industry on a whole in this country.

**Keywords:** berries; Borovička; juniper; α-pinene; raw material; essential oil; weight





## 1. Introduction

*Juniperus* genus has about 60 species of evergreen trees or shrubs, widely dispersed in the Northern Hemisphere, mostly in Eastern parts [1]. Juniper berries of various species have been used for medicinal purposes since antiquity. The berries of several species are consumed as a food by birds. Humans should not consume the berries or oil of different juniper species as some of them may be toxic; the medicinal use of small amounts of oil of some species of juniper has occasionally resulted in death [2,3]. Many of the species are grown as ornamental plants and some are also harvested for timber [4].

Common juniper (*Juniperus communis* L.) is the most widespread of the juniper species; it is indigenous to Eurasia and North America. As a species with a broad ecological amplitude, common juniper is spread across all of Slovakia with an altitude extending from the lowlands up to 1495 m above sea level [5]. It is a shrub or small tree which

is 2–20 ft. high and with reddish brown bark which shreds off in papery peels. Leaves (needles) taper to a spiny tip, in whorls of threes with two white bands above (or one white band sometimes divided by a green midrib with a broader green margin). It is a dioecious plant species.

Strictly, a "berry" is a fleshy or pulpy fruit derived from the ovary of a single flower, and the term is inappropriate for the junipers since the fleshy structure containing the seed is derived from three to six fleshy cone scales, each one-ovuled; however, the term berry is universally accepted [6].

Small cones, some male, others female, develop in early summer at the base of a few of the needle-like leaves. Male cones are yellow, while female cones are hard, pungent, and bluish-green with a whitish, waxy coating. The common juniper is an anemophilous species, but a certain degree of entomophily is also quite common. Female cones become fleshy berries, requiring 2 years to ripen, and turning gradually from green to blue-black. Berries are grown on short stalk, round to broadly oval, bluish black, usually with three seeds [7,8].

Juniper berries are one of the most widely used herbal diuretics. Approved in Germany, juniper tea is used for digestive problems and also to stimulate appetite. Their anti-inflammatory and spasm-reducing activity has been confirmed, and it may also contribute to diuretic activity. Fruits eaten fresh or in tea are a folk remedy used as a diuretic and urinary antiseptic for cystitis, carminative for flatulence, antiseptic for intestinal infections, once used for colic, coughs, stomach aches, colds, and bronchitis. Externally, they are used for sores, aches, rheumatism, arthritis, snakebites, and cancer. The content of volatile oil is responsible for diuretic and intestinal antiseptic activity. Diuretic activity results from the irritation of renal tissue [9,10].

Juniper fruits are often fermented for the production of alcoholic beverages; the fruits form the raw material for the production of the Slovak national drink "Borovička". The "brother" of this beverage is the Anglo-Saxon "Gin", which has been popular in the Western World for over 300 years [11]. Berries, fresh or dried, are sometimes used directly as a food, but they are crushed before use. The smell is bittersweet, very reminiscent of gin, pine, and turpentine. The taste is sweet, slightly pungent, with a bitter aftertaste [12,13].

The aim of this study is the monitoring of the habitats of common juniper (*Juniperus communis* L.) in the territory of Slovakia with an emphasis on the determination of natural components. Despite the economic importance of juniper populations, little is currently known about the extent and nature of the essential oil variability and composition. The comprehensive research presents variations in the yield of juniper essential oils and their qualitative–quantitative characteristics in relation to different chemotypes.

## 2. Material and Methods

### 2.1. Collection of the Plant Material

The fruits were collected from 25 localities in Slovakia [14] from 2018 to 2020 (Table 1, Figure 1). The biotope consists of open to dense stands of common juniper together with other light-loving species of trees, mainly shrubs, which occur within grass–herb or shrub vegetation communities. Juniper most often spreads on extensively used pastures, and thanks to its sharp needles, it is not threatened by grazing. On the contrary, it has a competitive advantage, because plants typically grow in rocky, infertile soils in fields, meadows, pastures, open woods, and other settings, almost from sea level to Alpine sites.

The preparation of samples after their collection was conducted by cleaning, de-needling, sorting the samples into groups, and drying at a temperature of 38 °C for 6 h in a dryer. The drying process evaporates excess water, and the final moisture content of the plant material for further processing was about 12%. The next step was to determine the weight of 100 juniper berries; this operation was repeated 6 times for each sample. Measurements were made on Sartorius analytical scale of the CPA type.

The fruits were separated and dried in a sheltered, open-air area at a temperature below 32 °C for 14 to 20 days with low humidity of 2 to 5%. The moisture content of the

fruit tissue was lowered to 12% to prevent mould infection. The dried materials were cleaned, packed, labelled, and stored in a clean and dry place until further use.

### 2.2. Isolation of Juniper Essential Oil

Each sample of dried fruits with a weight of 20 g was ground in a blender. The essential oil from this raw material was prepared using hydro-distillation (2 h) in Clevenger-type apparatus according to the European Pharmacopoeia [15]. Hexane was used as an extraction agent. The extracted essential oils were stored under $N_2$ atmosphere at +4 °C in a dark place before their GC-MS analyses.

### 2.3. GC/MS Analysis

The main components of the essential oil were determined using a GC-MSD system on a Varian 3090/MS Saturn 2100 T instrument with Split–Splitless injection inlet, injection volume: 2 µL, MSD detector, RX-5MS column, 30 m × 0.25 mm i. d., with internal diameter: 0.25 µm, carrier gas: helium (21 p.s.i.) with a flow rate of 1.50 mL/min BPX-5, 50 m long with an internal diameter of 0.25 mm and with a stationary phase thickness of 0.25 µm. Temperature program: 50 °C = 0 min.; 3 °C/min to 250 °C; 250 °C = 15 min.

**Table 1.** The basic characters of selected sites in Slovakia with the occurrence of juniper plants.

| | Locality | Geographical Latitude | Terrestrial Longitude | Altitude [m] | Aspect | Slope |
|---|---|---|---|---|---|---|
| 01 | Kišovce—Hôrka | N 49°02′84″ | E 20°38′11″ | 620 | northeast | 30° |
| 02 | Liptovská Lužná 2 | N 48°56′30″ | E 19°19′12″ | 700 | south | 15° |
| 03 | Liptovská Lužná 1 | N 48°56′31″ | E 19°19′15″ | 730 | southwest | 25° |
| 04 | Selčianske sedlo | N 48°45′53″ | E 19°12′24″ | 380 | northeast | 10° |
| 05 | Kráľová 1 | N 48°52′58″ | E 20°08′21″ | 1272 | east | 35° |
| 06 | Liptovská Lužná 3 | N 48°56′29″ | E 19°19′13″ | 710 | southeast | 35° |
| 07 | Vihorlat—Užok | N 48°59′05″ | E 22°51′53″ | 380 | southwest | 12° |
| 08 | Horné lazy 2 | N 48°48′51″ | E 19°36′00″ | 475 | northeast | 14° |
| 09 | Kráľová 2 | N 48°52′51″ | E 20°08′25″ | 1010 | south | 25° |
| 10 | Donovaly | N 48°52′48″ | E 19°13′21″ | 960 | north | 35° |
| 11 | Cerovo | N 48°15′17″ | E 19°09′26″ | 468 | northwest | 21° |
| 12 | Iliaš 1 | N 48°41′55″ | E 19°18′37″ | 340 | northeast | 12° |
| 13 | Priechod—West | N 48°46′45″ | E 19°13′54″ | 480 | west | 20° |
| 14 | Priechod—South | N 48°46′41″ | E 19°13′49″ | 420 | south | 15° |
| 15 | Spišský hrad | N 49°00′00″ | E 20°46′06″ | 628 | south | 30° |
| 16 | Priechod—East | N 48°46′40″ | E 19°13′50″ | 390 | east | 10° |
| 17 | Karpaty—Turka | N 49°09′27″ | E 23°01′21″ | 745 | southwest | 15° |
| 18 | Horné lazy | N 48°48′51″ | E 19°36′50″ | 520 | northeast | 30° |
| 19 | Lackov | N 48°19′15″ | E 19°11′12″ | 476 | southeast | 16° |
| 20 | Pravica | N 48°19′01″ | E 19°27′26″ | 356 | southwest | 12° |
| 21 | Chrámec—Teplá dolina 1 | N 48°15′35″ | E 20°10′56″ | 248 | northeast | 22° |
| 22 | Iliaš 2 | N 48°41′52″ | E 19°18′32″ | 320 | east | 16° |
| 23 | Chrámec—Vlčia dolina 3 | N 48°16′24″ | E 20°10′54″ | 250 | east | 10° |
| 24 | Chrámec—Vlčia dolina 4 | N 48°16′18″ | E 20°10′45″ | 220 | south | 10° |
| 25 | Chrámec—Teplá dolina 2 | N 48°15′34″ | E 20°10′60″ | 196 | south | 5° |

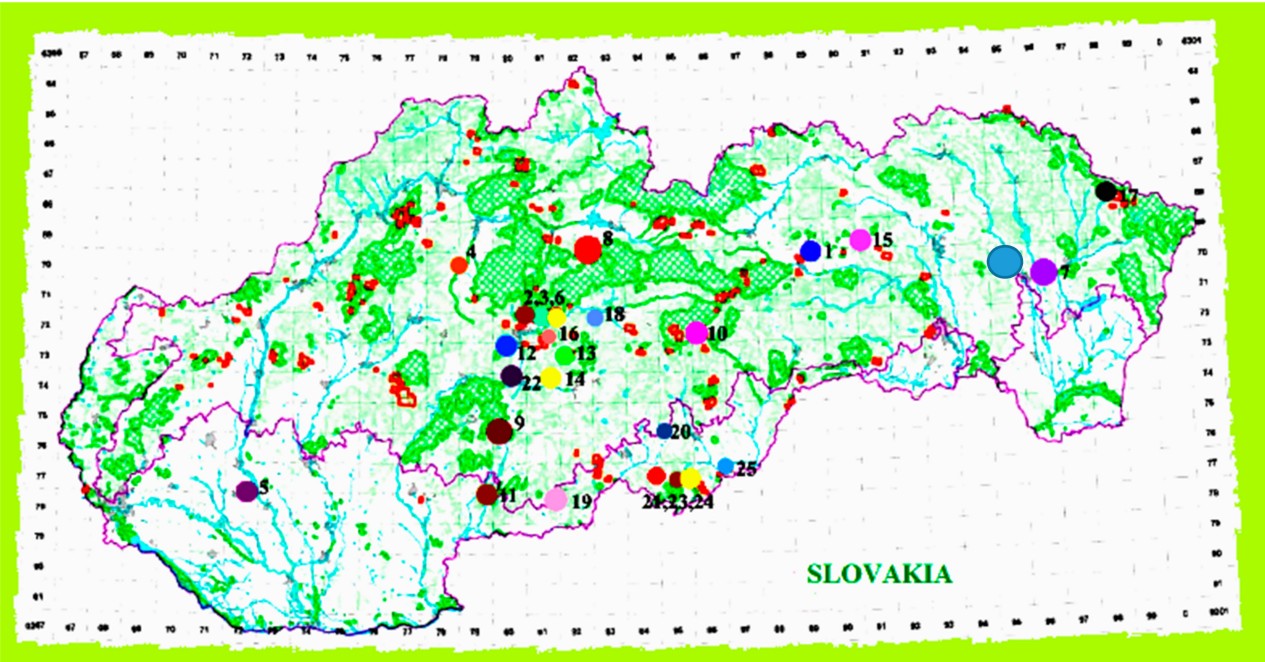

**Figure 1.** Slovakia—the localities [16] where juniper berries were collected: 1. Kišovce—Hôrka, 2. Liptovská Lužná 2, 3. Liptovská Lužná 1, 4. Selčianske sedlo, 5. Kráľová 1, 6. Liptovská Lužná 3, 7. Vihorlat—Užok, 8. Horné lazy 2, 9. Kráľová 2, 10. Donovaly, 11. Cerovo, 12. Iliaš 1, 13. Priechod—West, 14. Priechod—South, 15. Spišský hrad, 16. Priechod—East, 17. Karpaty—Turka, 18. Horné lazy, 19. Lackov, 20. Pravica, 21. Chrámec—Teplá dolina 1, 22. Iliaš 2, 23. Chrámec—Vlčia dolina 3, 24. Chrámec—Vlčia dolina 4, 25. Chrámec—Teplá dolina 2.

The identification of the individual components of the essential oil was conducted using the retention times of 40 authentic standards of the components supplied by the companies Extrasynthesis, Merck, Fulka, and Sigma-Aldrich and Kovats indexes (used $C_5$–$C_{22}$ alkanes), and the components were then integrated in NIST 98 library. The spectra of the individual components of the essential oil were compared with the mass spectra using the literature [17,18].

Components were identified using their GC retention times, and the resulting values were comparable to those of the literature. Oil component standards for comparison were supplied by Extrasynthese, s.a., Genay, France.

*2.4. Statistical Analysis*

Several statistical methods and biometric parameters of the obtained data were used: arithmetic means, standard deviations, Student *t*-test at the 0.05% level of significance (*n* = 6), and exploratory analysis with a graphic representation of the frequencies of selected quantitative features using variation curves [19]. Statistica Program V12, which was used for processing the measured data, is well-known statistical software developed for mass processing and evaluation of data with a subsequent tabular or graphic presentation. For the biostatistical evaluation of the obtained results, the following statistical tests were used for 130 data (*n* = 25). The Mann–Whitney U-test was used to compare the levels of the substances from two locations (Kisovce 01, Liptovská Lužná 2 02). Due to the low number of samples from the sites, there is a more appropriate choice when detecting significant differences for sites in chemotype D. The Kruskal–Wallis test is a non-parametric analysis of variance that was used when comparing levels from more than 3 sites (for each chemotype separately). The choice of method was made due to the low number of samples from individual locations (5 each). With this method, we can find out if there is a difference in level between any sample pair. The following multiple comparison test is able to specify significant differences between pairs. Statistical analysis was performed using confidence

intervals with a significance level $p < 0.05$ using calculation through the mean, standard deviation, and standard error.

The differences between juniper populations for the mean values of the essential oil constituents were detected using an ANOVA. All statistical data were calculated using SAS JMP Statistical Discovery, and after that a dendrogram (ward method) and a relationship diagram (*Juniperus populations* × the essential oil constituents) were made [20].

## 3. Results

### 3.1. Fruit Weights and Essential Oil Yield

For a long time, female plants with juniper fruits have been interesting for their harvesting. Common juniper is a highly variable species occurring in several forms (Figure 2), including a small tree up to 0.1 m pyramidal or columnar in shape with an upright bush, even a prostatic bush forming a mat. According to calculated variability in the morphological characteristics of trees and shrubs, different total amounts of harvested fruits were obtained, and, accordingly, significant differences in their size (Figure 3) and weight (Table 2) were proven. These findings prove that the entire spectrum of abiotic and biotic factors affects the ontogenetic development of plants and their populations in individual locations in Slovakia.

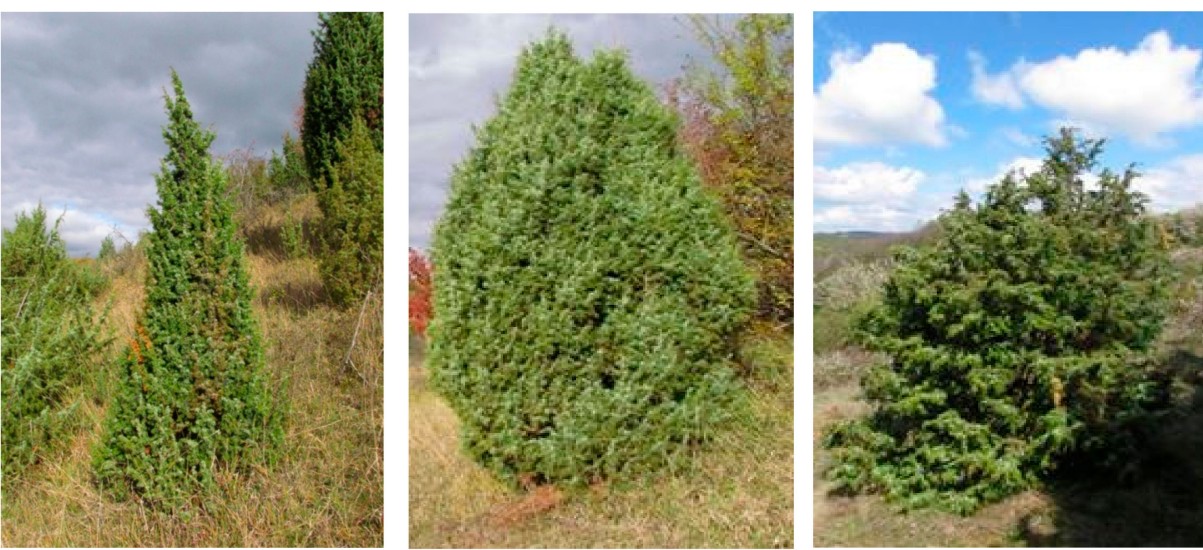

**Figure 2.** Tree forms of juniper bushes (pyramidal, ovoid, and spreading shrub) grown in Slovak montane and sub montane areas.

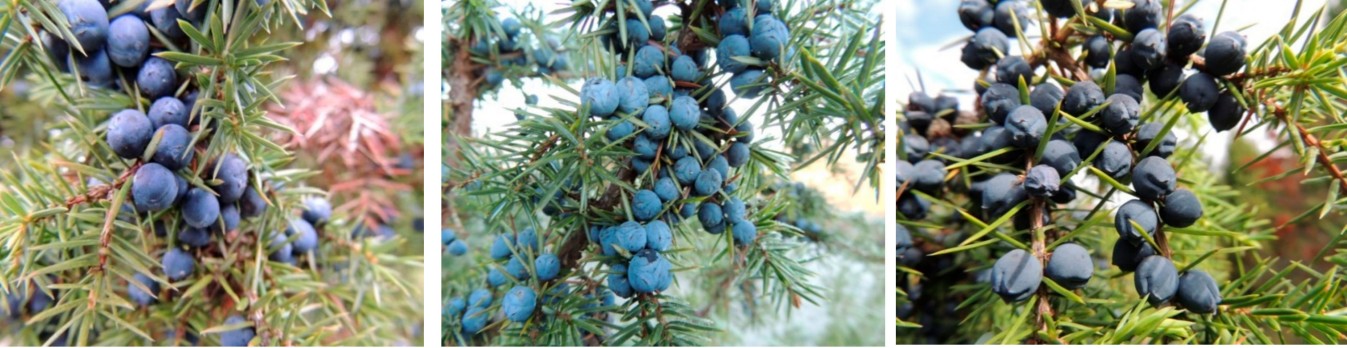

**Figure 3.** Shapes of juniper berries—from small to large, from round to elongated, to spindle-shaped.

**Table 2.** Berry weights (100 pieces/pcs/) and yield of juniper essential oil in % (*v/w*—expressed as a dry weight).

|  | Habitat/Locality | Berry Weights [g] | Essential Oil Yield [%] |
|---|---|---|---|
| 01 | Kišovce—Hôrka | 5.84 ± 0.29 | 0.5 ± 0.05 |
| 02 | Liptovská Lužná 2 | 5.84 ± 0.16 | 0.6 ± 0.03 |
| 03 | Liptovská Lužná 1 | 6.27 ± 0.37 | 0.6 ± 0.12 |
| 04 | Selčianske sedlo | 6.58 ± 0.07 | 0.6 ± 0.11 |
| 05 | Kráľová 1 | 6.59 ± 0.09 | 0.6 ± 0.09 |
| 06 | Liptovská Lužná 3 | 6.87 ± 0.08 | 0.7 ± 0.04 |
| 07 | Vihorlat—Užok | 7.05 ± 0.47 | 0.6 ± 0.15 |
| 08 | Horné lazy 2 | 7.22 ± 0.04 | 0.7 ± 0.06 |
| 09 | Kráľová 2 | 7.43 ± 0.19 | 0.9 ± 0.12 |
| 10 | Donovaly | 7.53 ± 0.21 | 0.9 ± 0.11 |
| 11 | Cerovo | 7.83 ± 0.26 | 0.9 ± 0.10 |
| 12 | Iliaš 1 | 8.07 ± 0.13 | 0.9 ± 0.08 |
| 13 | Priechod—West | 8.09 ± 0.17 | 0.9 ± 0.09 |
| 14 | Priechod—South | 8.33 ± 0.23 | 1.1 ± 0.04 |
| 15 | Spišský hrad | 8.44 ± 0.23 | 1.2 ± 0.07 |
| 16 | Priechod—East | 8.69 ± 0.20 | 1.2 ± 0.10 |
| 17 | Karpaty—Turka | 8.89 ± 0.28 | 1.2 ± 0.16 |
| 18 | Horné lazy 1 | 9.18 ± 0.08 | 1.2 ± 0.14 |
| 19 | Lackov | 9.24 ± 0.08 | 1.2 ± 0.12 |
| 20 | Pravica | 9.43 ± 0.19 | 1.4 ± 0.11 |
| 21 | Chrámec Teplá dolina 1 | 10.35 ± 0.16 | 1.5 ± 0.08 |
| 22 | Iliaš 2 | 11.65 ± 0.10 | 1.5 ± 0.11 |
| 23 | Chrámec Vlčia dolina 3 | 14.04 ± 0.27 | 1.5 ± 0.05 |
| 24 | Chrámec Vlčia dolina 4 | 14.64 ± 0.24 | 1.6 ± 0.07 |
| 25 | Chrámec Teplá dolina 2 | 15.08 ± 0.34 | 1.8 ± 0.06 |

From the measured values, we found that the samples taken from the Protected Landscape Area—Cerová vrchovina sites had the highest values of weight, where their values ranged from 10.35 ± 0.16 g/100 pcs to 15.08 ± 0.34 g/100 pcs of fruits. The lowest weight values were found in the locations of Poprad—Kišovce Hôrka and Liptovská Lužná 2. In Kišovce, the value reached 5.84 g/100 juniper berries.

The content of juniper essential oil isolated from all dry fruit samples ranging from 0.5 ± 0.05% to 1.8 ± 0.10% (Table 2). The measured values showed that the samples taken from the sites of the PLA Cerová vrchovina, Chrámec Teplá dolina 2 had the highest weights, where the amount of essential oil was 1.8 ± 0.1%. The lowest amount of essential oil, 0.5 ± 0.1%, was found in juniper fruits that were collected at the location Poprad—Kišovce Hôrka.

The box plot of the amounts of juniper essential oil from 25 locations in Slovakia (Figure 4) illustrates the individual amounts of isolated essential oil as a percentage. Individual localities have significant differences in the quantity of essential oil. The mentioned detection documents the importance of site selection for the quantities and quality of the harvest of plant material for the distillery industry. There was up to 1% of essential oil in fruits, the largest number of locations, i.e., 15. There were only five locations with content above 1.5%, which are located up to 200 m above sea level, with the exception of the

Iliaš 2 location, which is located in the valley of the Hron River in Banská Bystrica. The largest quantities of essential oil were clearly isolated from locations in southern Slovakia, where the altitude is up to 150 m a.s.l. and warm to hot weather prevails during the growing season, with maximum average temperatures of 26 °C during the months of July and August.

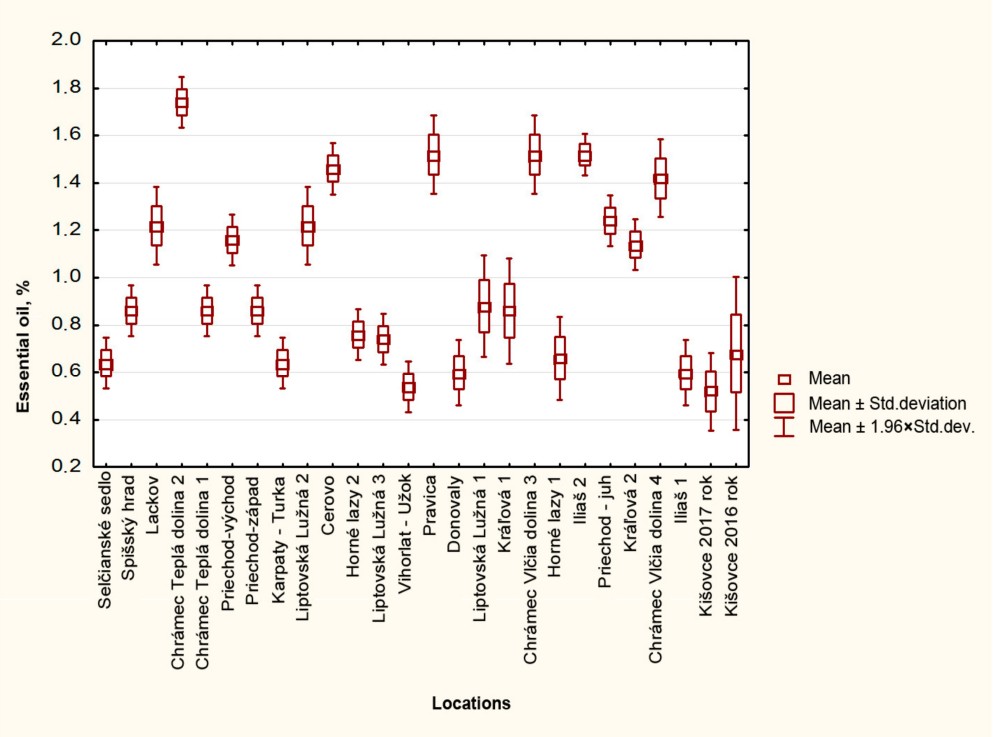

**Figure 4.** Box graph of quantities of juniper essential oil obtained from locations in Slovakia.

The oil was distilled from dry black, fully ripe berries. It is transparent, fluid, and colourless with a tinge of greenish-yellow. The aroma is similar to that of pine, but more peppery, hot, and balsamic, with a burning, somewhat bitter taste. In accordance with the distillation process, the selected physical parameters of the juniper essential oil were determined (Table 3). The obtained results of the following parameters—bulk density, optical activity, index light refraction, and ash determination—confirmed the findings that the observed parameters follow pharmacopoeia limits.

**Table 3.** Selected physical and parameters of juniper essential oil.

| Physical and Chemical Parameters | Technical Parameters [9] | Determined Parameters |
| --- | --- | --- |
| Bulk density | 0.860–0.879 g/cm$^3$ | 0.876 ± 0.015 g/cm$^3$ |
| Optical activity | −10°−−30° | −12 ± 1° |
| Index light refraction | 1.457–1.467 | 1.464 ± 0.001 |
| Ash | max. 0.5% | <0.5% |

### 3.2. The Dependence of Essential Juniper Oil Yield on Fruit Weights

Correlation analysis statistically proved the dependence at the significance level of 0.05 that the size of the fruit affects the amount of essential oil (Figure 5), with higher essential oil contents concentrated in the southern regions of Slovakia. It is interesting, however, that, historically, in this part of the country; an alcoholic drink of the pine type has never been produced, either officially or as a homemade product.

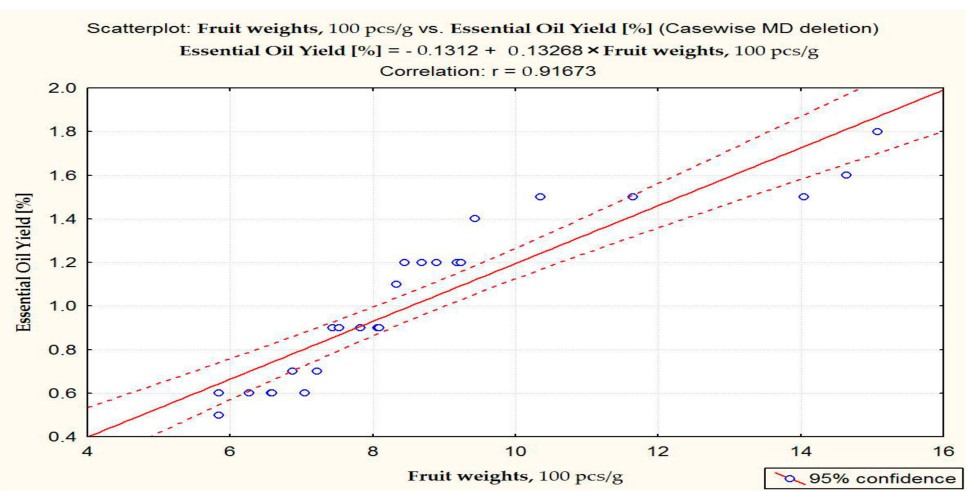

**Figure 5.** Correlation dependence of the amount of essential oil on fruit weight.

*3.3. Essential Oil and Its Qualitative–Quantitative Characteristics*

Using GC/MS analyses, 25 different chemical components were identified in the essential oil isolated from juniper berries. For the determination of individual chemical types, only the main siliceous components were selected (Table 4), which are important not only for medicinal utilisation but also for industrial processing in the distillery. Aromatic α-pinene was identified as the first main substance, sabinene, and β-myrcene followed according to the amount in the essential oil.

**Table 4.** Main components of the essential oil isolated from juniper berries belonging to east and central Slovakia.

| | Localities | Basic Composition of Essential Oil in % | | | | | | |
|---|---|---|---|---|---|---|---|---|
| | | α-Pinene | Sabinene | β-Pinene | β-Myrcene | Limonene | Boranyl acetate | β-Caryophylene |
| 01 | Kišovce—Hôrka | 61.0 ± 0.60 | 3.5 ± 0.30 | 2.5 ± 0.40 | 10.0 ± 0.30 | 3.0 ± 0.45 | ≤0.3 | 3.5 ± 0.55 |
| 02 | Liptovská Lužná 2 | 37.0 ± 0.86 | 14.5 ± 1.30 | 1.5 ± 0.41 | 10.0 ± 0.43 | 2.0 ± 0.22 | ≤0.2 | 9.5 ± 0.45 |
| 03 | Liptovská Lužná 1 | 46.0 ± 0.60 | 10.5 ± 0.68 | 2.0 ± 0.12 | 14.0 ± 0.30 | 3.1 ± 0.35 | ≤0.3 | 6.0 ± 0.45 |
| 04 | Selčianske sedlo | 29.0 ± 0.68 | 12.5 ± 0.68 | 2.5 ± 0.30 | 12.0 ± 0.68 | 2.5 ± 0.21 | ≤0.3 | 6.5 ± 0.30 |
| 05 | Kráľová 1 | 46.0 ± 1.03 | 13.0 ± 0.30 | 2.0 ± 0.17 | 11.0 ± 0.35 | 2.2 ± 0.31 | ≤1.0 | 12.0 ± 2.05 |
| 06 | Liptovská Lužná 3 | 41.0 ± 0.60 | 14.5 ± 1.30 | 3.0 ± 0.25 | 13.0 ± 0.30 | 3.0 ± 0.21 | ≤0.3 | 8.0 ± 0.25 |
| 07 | Vihorlat—Užok | 44.1 ± 0.08 | 9.50 ± 0.30 | 2.5 ± 0.12 | 14.2 ± 0.16 | 4.50 ± 1.31 | ≤0.7 | 7.9 ± 0.30 |
| 08 | Horné lazy 2 | 37.0 ± 0.86 | 9.5 ± 0.30 | 2.7 ± 0.35 | 10.0 ± 0.30 | 2.0 ± 0.45 | ≤0.5 | 9.0 ± 1.00 |
| 09 | Kráľová 2 | 46.0 ± 0.68 | 9.5 ± 0.30 | 2.9 ± 0.25 | 16.0 ± 0.66 | 2.1 ± 0.15 | ≤0.5 | 8.7 ± 0.75 |
| 10 | Donovaly | 44.0 ± 0.86 | 8.0 ± 0.30 | 2.0 ± 0.09 | 9.0 ± 0.30 | 6.0 ± 0.15 | ≤0.5 | 8.5 ± 0.45 |
| 11 | Cerovo | 36.0 ± 0.61 | 9.5 ± 0.30 | 2.5 ± 0.20 | 16.0 ± 0.43 | 4.0 ± 0.55 | ≤0.3 | 5.5 ± 0.50 |
| 12 | Iliaš 1 | 49.0 ± 1.91 | 8.0 ± 0.42 | 2.2 ± 0.06 | 11.0 ± 0.30 | 3.2 ± 0.21 | ≤1.0 | 6.1 ± 0.35 |
| 13 | Priechod—West | 38.0 ± 0.68 | 15.0 ± 0.43 | 3.0 ± 0.27 | 16.0 ± 0.43 | 2.5 ± 0.10 | ≤0.5 | 10.0 ± 1.50 |
| 14 | Priechod—South | 47.0 ± 0.86 | 12.5 ± 0.30 | 1.9 ± 0.35 | 11.0 ± 0.42 | 1.7 ± 0.15 | ≤0.3 | 9.1 ± 0.45 |
| 15 | Spišský hrad | 32.0 ± 0.86 | 20.5 ± 0.68 | 2.5 ± 0.10 | 8.0 ± 0.43 | 7.5 ± 0.15 | ≤0.5 | 5.5 ± 0.25 |
| 16 | Priechod—East | 37.0 ± 1.35 | 10.5 ± 0.68 | 3.0 ± 0.15 | 9.0 ± 0.43 | 3.5 ± 0.55 | ≤0.5 | 17.0 ± 1.05 |
| 17 | Karpaty—Turka | 47.0 ± 1.52 | 9.21 ± 0.42 | 1.42 ± 0.19 | 13.50 ± 0.95 | 2.40 ± 1.50 | ≤0.2 | 11.5 ± 1.30 |
| 18 | Horné lazy 1 | 48.0 ± 0.60 | 10.0 ± 0.30 | 2.0 ± 0.15 | 15.0 ± 0.30 | 2.7 ± 0.31 | ≤0.3 | 4.3 ± 0.15 |
| 19 | Lackov | 32.0 ± 1.09 | 13.5 ± 0.68 | 2.4 ± 0.42 | 20.0 ± 0.96 | 3.5 ± 0.25 | ≤0.5 | 8.5 ± 0.50 |
| 20 | Pravica | 43.0 ± 1.35 | 12.0 ± 0.96 | 1.0 ± 0.06 | 14.0 ± 0.30 | 2.0 ± 0.15 | ≤0.5 | 8.2 ± 0.49 |

Table 4. *Cont.*

| Localities | | Basic Composition of Essential Oil in % | | | | | | |
|---|---|---|---|---|---|---|---|---|
| | | α-Pinene | Sabinene | β-Pinene | β-Myrcene | Limonene | Boranyl acetate | β-Caryophylene |
| 21 | Chrámec, Teplá dolina 1 | 37.0 ± 0.68 | 19.5 ± 0.68 | 2.0 ± 0.15 | 13.0 ± 0.30 | 2.0 ± 0.31 | ≤0.3 | 4.5 ± 0.22 |
| 22 | Iliaš 2 | 47.0 ± 0.60 | 14.5 ± 0.68 | 1.5 ± 0.12 | 16.0 ± 0.30 | 0.7 ± 0.09 | ≤0.3 | 3.3 ± 0.18 |
| 23 | Chrámec, Vlčia dolina 3 | 45.0 ± 0.86 | 16.0 ± 0.42 | 2.6 ± 0.22 | 9.0 ± 0.30 | 4.7 ± 0.25 | ≤0.5 | 3.1 ± 0.15 |
| 24 | Chrámec, Vlčia dolina 4 | 49.0 ± 1.35 | 22.0 ± 0.96 | 2.0 ± 0.15 | 2.0 ± 0,30 | 5.1 ± 0.15 | ≤0.5 | 3.0 ± 0.25 |
| 25 | Chrámec, Teplá dolina 2 | 37.0 ± 0.68 | 20.0 ± 0.43 | 3.0 ± 0.25 | 7.0 ± 0.30 | 1.5 ± 0.11 | ≤0.3 | 9.0 ± 0.72 |

### 3.3.1. α-Pinene Content

α-Pinene is an organic compound of the terpene class, one of the two isomers of pinene. It is an alkene that contains an active four-membered ring. It is found in the oils of many conifer species, including juniper (*Juniperus communis* L.).

The content of aromatic α-pinene in the essential oil varies from 29.0 ± 0.68% at the Selčianske sedlo location to 41.0 ± 0.60% at the Liptovská lužná location, and from 50.0 ± 0.60% at the Iliaš location up to 61.0 ± 0.60% at the Kišovce-Hôrka location. These are locations located in both eastern and central Slovakia (Figure 6).

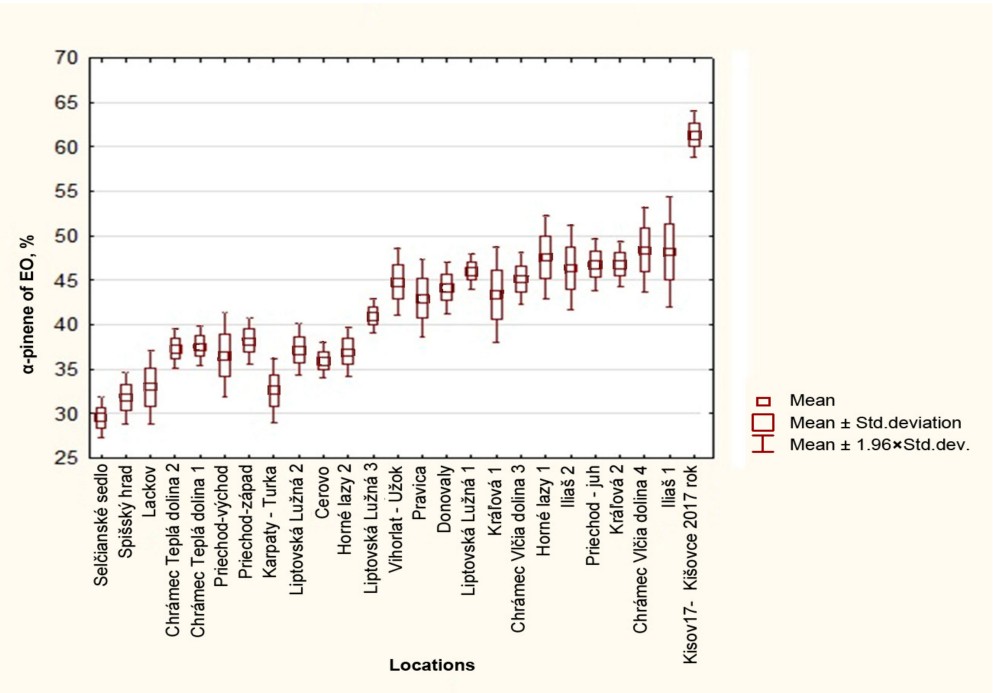

**Figure 6.** Box graph of quantities of α-pinene essential oil from localities in Slovakia.

These findings proved that the tolerance of Slovak juniper populations in connection with the content of α-pinene in essential oil is significant. It means that the ecological amplitude and thus the adaptation of plants to the specific properties of the biotope, such as temperature, salinity, and nitrogen content in the soil, are wide with considerable ecotolerance. The adaptation of plants to climate conditions is usually in close connection to the entire average of the ecological amplitude of the species in correlation towards the existing conditions.

### 3.3.2. Sabinene Content

From the point of view of the quantitative amount, the second natural substance in essential oil is the monotropic hydrocarbon sabinene. Chromatographic records confirmed that its quantity interval ranged from 3.5 ± 0.30% at the Kišovce—Hôrka location through

to 11.0 ± 0.42% at the Liptovská Lužná location, and up to 22.0 ± 0.96% at locality Chrámec-Vlčia dolina (Figure 7).

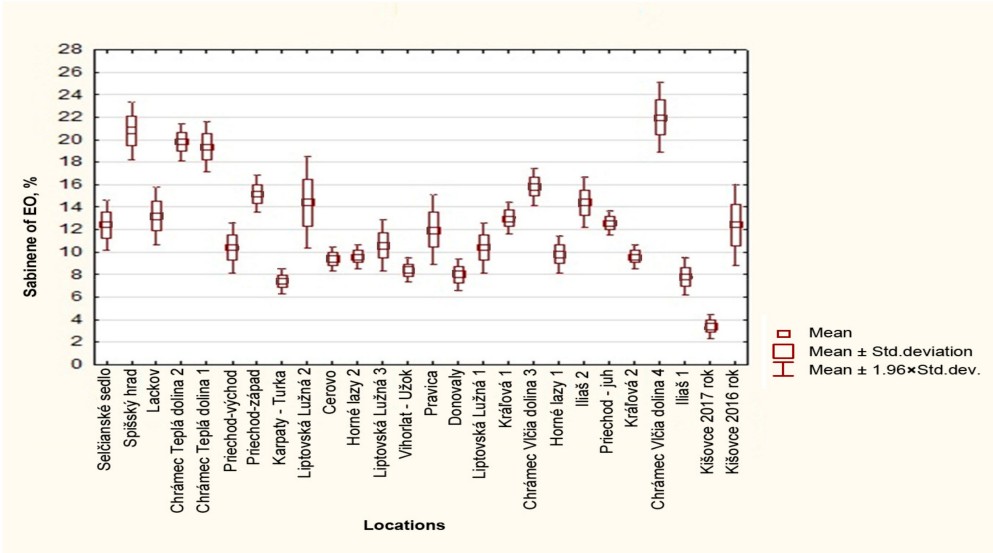

**Figure 7.** Box graph of quantities of sabinene essential oil according to localities in Slovakia.

### 3.3.3. β-Myrcene Content

As the third main monoterpene component, β-myrcene was determined. Its content ranged from 2 ± 0.30% at the Chrámec Vlčia dolina location through to 12.0 ± 0.68% at the Selčianske sedlo location, and up to 20 ± 0.96% at the Lackov location in the essential oil isolated from Slovakian juniper berries (Figure 8).

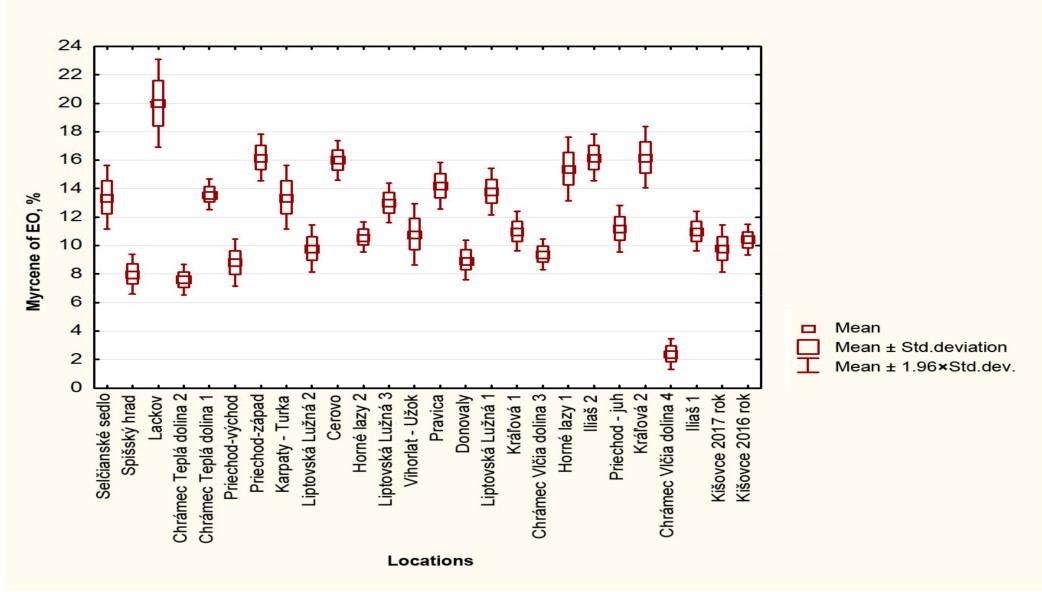

**Figure 8.** Box graphs of β-myrcene content in the essential oil of juniper berries from localities in Slovakia.

Table 5 presents the qualitative–quantitative characteristics of complete GC-MS analyses of juniper fruit essential oils from three different locations in Slovakia. The interpretation of these results is interesting from several aspects. The interval of α-pinene content ranges from the highest content of 61.00 ± 0.60% (locality: Kišovce-Hôrka) to the lowest 37.60 ± 2.23% (locality: Chrámec—Vlčia dolina). There is a very large difference in the content of sabinene as its content in the essential oil from the locality Kišovce-Hôrka was

3.50 ± 0.30% and for the other localities (Chrámec—Teplá dolina and Chrámec—Vlčia dolina) it was over 19.00%. The myrcene content showed no statistical differences at all three mentioned locations. A significant difference in the content of essential oil was also noted for α-caryophyllene. In the connection with the overall results of the essential oil content, it is confirmed that the natural plant species of common juniper occurs in a wide range of environmental factors and their influence. This species has wide ecological amplitude, and we refer to it as indifferent in certain conditions. The biotopes of the juniper population are of great importance in forestry because they reliably indicate the properties of the locality, the potential of the plant community, and its production potential.

**Table 5.** The marked differences in chemical composition of juniper essential oil from tree localities in Slovakia [%].

| Component as % of Essential Oil * | GC-MS $t_R$ (min.) ** | Localities of Juniper Fruit Collection | | |
|---|---|---|---|---|
| | | Kišovce—Hôrka | Chrámec—Teplá Dolina | Chrámec—Vlčia Dolina |
| α-thujone | 8.04 | trace | trace | trace |
| α-pinene | 8.16 | 61.00 ± 0.60 | 46.50 ± 3.35 | 37.60 ± 2.23 |
| β-pinene | 13.27 | 2.50 ± 0.30 | 2.51 ± 0.92 | 2.57 ± 0.92 |
| sabinene | 14.31 | 3.50 ± 0.30 | 19.67 ± 2.08 | 19.77 ± 1.67 |
| β-phellandrene | 16.12 | 0.34 ± 0.03 | 0.07 ± 0.02 | 0.08 ± 0.02 |
| camphene | 16.49 | 0.22 ± 0.05 | 0.24 ± 0.06 | 0.26 ± 0.05 |
| β-myrcene | 17.13 | 10.01 ± 0.30 | 8.03 ± 2.02 | 10.56 ± 0.05 |
| limonene | 19.57 | 2.50 ± 0.30 | 5.06 ± 0.92 | 2.27 ± 1.14 |
| γ-terpinene | 19.62 | 0.85 ± 0.06 | 0.31 ± 0.06 | 0.35 ± 0.07 |
| cis-sabinene hydrate | 20.43 | trace | trace | trace |
| terpinolene | 23.76 | 0.33 ± 0.08 | 0.37 ± 0.05 | 0.40 ± 0.06 |
| trans-sabinene hydrate | 24.09 | trace | trace | trace |
| 1-terpinene-4-ol | 24.44 | 0.12 ± 0.02 | 0.95 ± 0.09 | 1.56 ± 0.92 |
| β-elemene | 38.03 | 0.05 ± 0.01 | 0.23 ± 0.01 | 0.43 ± 0.02 |
| methyl citronelate | 38.75 | trace | trace | trace |
| boranyl acetate | 38.97 | 0.10 ± 0.01 | 0.43 ± 0.09 | 0.40 ± 0.09 |
| γ-elemene | 41.25 | 1.09 ± 0.06 | 1.25 ± 0.07 | 1.74 ± 0.47 |
| (Z)-β-farnesene | 43.33 | 1.33 ± 0.27 | 1.44 ± 0.29 | 1.52 ± 0.29 |
| α-caryophyllene | 45.19 | 3.50 ± 0.30 | 6.51 ± 2.01 | 5.01 ± 1.66 |
| germacrene D | 45.98 | 0.99 ± 0.17 | 2.42 ± 0.95 | 1.81 ± 0.96 |
| γ-elemene | 46.21 | trace | trace | trace |
| α-gurjunene | 47.59 | 1.41 ± 0.08 | 1.22 ± 0.97 | 2.05 ± 0.96 |
| γ-cadinene | 49.41 | 0.27 ± 0.07 | 1.04 ± 0.14 | 0.25 ± 0.05 |
| β-cadinene | 50.11 | 1.12 ± 0.13 | 0.27 ± 0.07 | 0.10 ± 0.03 |
| δ-cadinene | 50.22 | 0.44 ± 0.03 | 0.16 ± 0.04 | 0.21 ± 0.08 |
| elemole | 51.09 | trace | trace | trace |
| germacrene D-4-ol | 51.42 | trace | trace | trace |
| spathulenole | 52.35 | trace | trace | trace |
| shyobunole | 53.76 | trace | trace | trace |

**Table 5.** *Cont.*

| Component as % of Essential Oil * | GC-MS $t_R$ (min.) ** | Localities of Juniper Fruit Collection | | |
| --- | --- | --- | --- | --- |
| | | Kišovce— Hôrka | Chrámec— Teplá Dolina | Chrámec— Vlčia Dolina |
| abietadiene | *54.06* | trace | trace | trace |
| 4-epi-abietale | *55.87* | trace | trace | trace |
| abieta-7,13-diene-3-on | *56.22* | trace | trace | trace |
| Total [%] | | 100.00 | 100.00 | 100.00 |

Note: *—data are expressed as area in % of the 100.00% of all identified peaks; **—retention times.

### 3.4. Hierarchical Cluster Analysis

In order to provide additional insights into the chemotypes of juniper essential oils, a hierarchical cluster analysis based on the constituents was carried out. The dendrogram of the analysis is shown in Figure 9. Based on this analysis, there are four different confirmed chemotypes:

- Chemotype A, dominated by α-pinene (29–33%) > sabinene (14–22%) > myrcene (8–22%);
- Chemotype B, dominated by α-pinene (35–47%) > sabinene (9–21%) > myrcene (9–17%);
- Chemotype C, dominated by α-pinene (49–54%) > sabinene (8–24%) > myrcene (11–18%);
- Chemotype D, dominated by α-pinene (60–62%) > sabinene (3–4%) > myrcene (10–11%).

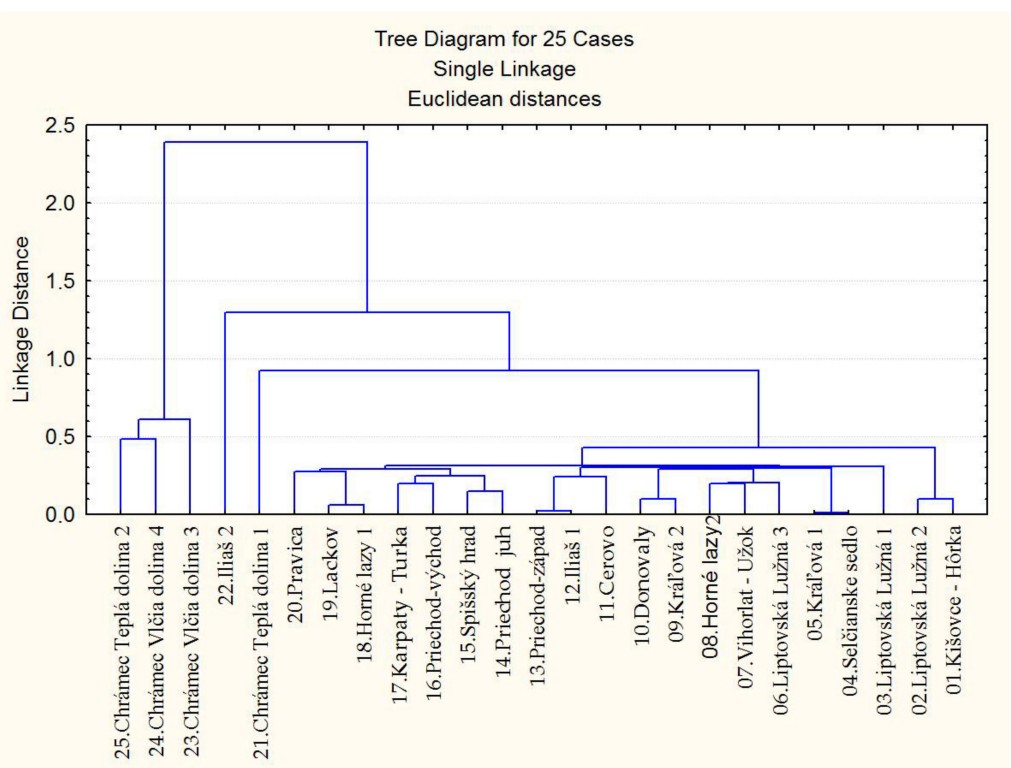

**Figure 9.** Dendrogram of relationships among juniper populations in Slovakia and their qualitative–quantitative characteristics of essential oils.

The results obtained from the cluster analysis show the existence of a high variability within the essential oils and differences among the following juniper populations collected

in different areas in Slovakia (Figure 9). From the 25 populations submitted to multivariate analysis, 4 well-defined groups of essential oils were differentiated using cluster analysis.

Four subclusters can be observed: the first subset contains tree populations collected in Chrámec—Teplá dolina 2, Chrámec—Vlčia dolina 4, and Chrámec—Vlčia dolina 3; the second subset includes the sample collected: Iliaš 2; the third subset includes the sample collected: Chrámec—Teplá dolina 1; the forth subset has twenty populations: Pravica, Lackov, Horné lazy 1, Karpaty—Turka, Priechod—East, Spišský hrad, Priechod South, Priechod—West, Iliaš 1, Cerovo, Donovaly, Kráľová 2, Horné lazy 2, Vihorlat—Užok, Liptovská Lužná 3, Kráľová 1, Selčianske sedlo, Liptovská Lužná 1, Liptovská Lužná 2, and Kišovce—Hôrka.

## 4. Discussion

Common juniper, *Juniperus communis* L., is a chamaephyte or phanerophyte depending upon growth form and habitat types, respectively [21,22]. In connection with that, the diameter of juniper berries ranged from 3 to 8 mm. From this detection, it is possible to conclude that the most fruits with the largest average size (6–7 mm) were collected from the locations Cerovo, Horné Lazy 2, Priechod—East [23]. In the case of the harvesting of junipers from plant populations in the Chrámec—Teplá dolina location, the diameter of berries was confirmed to be up to 6.5–9 mm, and only 5–7 mm at the Spiš castle on Ostrá hora. Significant differences in the average were also determined in their weight, when 100 juniper fruits in the first case weighed $15.08 \pm 0.34$ g versus $8.44 \pm 0.23$ g. Fruits harvested under the forest and near the village of Kišovce-Hôrka in the Tatra valley have a weight of only $5.84 \pm 0.19$ g/100 juniper berries.

Natural factors (such as soil structure, plant density, relief, sum of precipitation, and groundwater level) play an important role for the different structure and functions of juniper populations [24], which was confirmed by the significant differences in the weight of berries harvested in Slovakia. Similarly, as part of a harvesting expedition carried out in the region of the Stredna Gora mountain range, near the town of Kazalnak, in Bulgaria, fruits from two species were obtained: common juniper (*Juniperus communis* L.) and red juniper (*Juniperus oxycedrus* L.). The weight of 100 fruits of the first species was $10.19 \pm 0.14$ g, while the weight of 100 larger red juniper berries was $29.37 \pm 0.12$ g [25].

In cooperation with the laboratory Dr. Panghyova, interesting results were detected regarding the juniper berry carbohydrate contents and the amount of relevant essential oil. A higher content of essential oils was determined in fruits containing fewer carbohydrates. The contents of juniper essential oil isolated from all the dry fruit samples of all localities in Slovakia ranged within the interval from $0.50 \pm 0.10\%$ (416.7 g/kg carbohydrates) to $1.80 \pm 0.10\%$ (300.2 g/kg carbohydrates) [22].

Matovic et al. [26], in a research article from Serbia, published that the content of essential oil in juniper fruits varies from 2.13 to 3.25%. Another publication from 2011 [26] reported that the essential oil content of juniper berries harvested in a forest region ranged from 2.30 to 2.66%. Our results show that the essential oil yield ranged from $0.5 \pm 0.05\%$ to $1.8 \pm 0.10\%$. It has been confirmed that in the Balkan Peninsula, juniper berries contain higher amounts of essential oil. Comparing the yield of essential oil in different parts of Europe, it is important for the future development of the cultivation of this plant species to obtain enough raw material of a suitable quality for the distillery industry in Slovakia.

The number of published works focuses primarily on the qualitative–quantitative composition of isolated essential oil from fruits harvested from autochthonous populations of common juniper (*Juniperus communis* L.) in various areas of their occurrence. However, in most cases this analysis of essential oils content is linked to the biological properties of the medicinal plant, and the chemotype is not investigated [27,28].

The influence of the length of distillation on the amount of $\alpha$-pinene was monitored by Chatzopolou et al. [29] in Greece. It was concluded that the most suitable time period for juniper distillation is 1 h, when 27.8% of $\alpha$-pinene was isolated in the essential oil, for example, compared to 3 h distillation 22.4% and 6 h for only 9.7% of $\alpha$-pinene.

A study from Bulgaria reports on the essential oil composition of common juniper populations, while the predominant component was α-pinene with an interval of its content ranging from 21.4 to 38.4%, sabinene from 10.5 to 19.6%, limonene from 1.8 to 5.5%, and terpinen-4-ol from 3.2 to 7.5% [30]. As can be seen, the main component α-pinene predominates and presents a spectrum of other components (sabinene, limonene, terpinen-4-ol), while the important myrcene as well as β-pinene is almost missing [31].

Extensive research on the composition of essential oil isolated from needles (narrow, linear, and hard leaves) and unripe green berries was carried out in Lithuania [32,33]. Analysis using capillary gas chromatography as a higher content of β-pinene in the essential oil compared to α-pinene was shown. The analyses of the essential oil obtained from green juniper fruits did not confirm these conclusions, while the α-pinene content was up to $44.0 \pm 0.60\%$ and only $2.5 \pm 0.30\%$ for β-pinene, sabinene: $17.5 \pm 0.68\%$, myrcene: $5.0 \pm 0.31\%$, α-caryophyllene: $5.5 \pm 0.11\%$, and 1-terpinen-4-ol: $1.50 \pm 0.32\%$.

In relation to biological tests, Matovic et al. [34] confirmed the microbial effects of juniper from Serbia in protection against various pathogenic microorganisms (*Agrobacterium tumefaciens*, *Bacillus subtilis*, *Esherichia coli*, and *Pseudomonas flurescens*). They reported that the composition of the essential oil was 30.76% α-pinene, 19.37% sabinene, and 16.42% myrcene. In one of the Slovak samples from the Kišovce-Hôrka locality, there was a determined content of 61% α-pinene, 21% sabinene, and 18% myrcene (group D). At low exposure levels, α-pinene is a human bronchodilator and is highly bioavailable with 60% of absorption by human lungs with rapid metabolism and redistribution. In addition, this monoterpene has anti-inflammatory and antimicrobial effects.

In the last decade, chemotyping of common juniper (*Juniperus communis* L.) populations was done in Lithuania [35]. The essential oil content in the harvested juniper berry samples from 34 biotopes ranged from 0.4 to 2.9%. The most dominant chemotype was α-pinene; sabinene chemotype was proved only at three locations. The value of sabinene, as the main silica component, in these cases ranged from 34.1 to 40.8%, α-pinene from 11.7 to 27.8%, and myrcene from 4.3 to 12.8%. The Slovak α-pinene chemotypes contain 3.5 to 22.0% of sabinene in the essential oil. Juniper berries collected in the surroundings of the capital city of Lithuania, Vilnius, have high values of α-pinene contents (38.5–59.9%) in the essential oil, which are very similar to those in Slovakia (group D, location Kišovce-Hôrka, content of α-pinene: 29–61%). High values of α-pinene (53.6–62.3%) in juniper oil have been reported by Orav et al. [34] in berries harvested from stands at several locations in Estonia.

Juniper berry samples were harvested in Slovakia during 2012–2014 at five locations: Hôrka, Miľpoš, Lačnov, Zbojné, and Kamienka (parts of northeastern Slovakia) [36]. The percentage of essential oil varied from 0.6 to 1.9%; its α-pinene content ranged from 31.0 to 49.0%. The results confirmed the presence of a α-pinene chemotype. In the investigation, when studying 25 locations in Slovakia, the range of the content of essential oil was at a similar interval, but the values of α-pinene were in a much wider range. It was confirmed that the genetic diversity at the level of the common juniper population accurately characterised the chemotype diversity and its structure in Slovakia.

Fejer et al. [36] reported that the values for the β-myrcene content ranged from $7.00 \pm 0.5\%$ (Miľpoš) to $16.67 \pm 3.88\%$ (Hôrka), and for the essential oil component sabinene from $2.33 \pm 1.89\%$ (Hôrka) to $7.17 \pm 8.61\%$ (Miľpoš). In the investigation, for essential oil isolated from 25 locations, the amounts of myrcene were confirmed from $2.0 \pm 0.30\%$ (Chrámec Vlčia dolina 4) to $20.0 \pm 0.96\%$ (Lackov), and in the case of sabinene from $3.5 \pm 0.30\%$ (Kišovce-Hôrka) to $22.0 \pm 0.96\%$ (Chrámec—Vlčia dolina 4). This proved that the tolerance level of Slovak populations of the common juniper species in connection with the contents of the mentioned two components of the essential oil is extensive. Their ecological amplitude of biosynthesis and thus the adaptation of plants to the specific conditions of the given biotope is found with considerable eco-tolerance.

## 5. Conclusions

Juniper populations have undoubtedly declined many times in Slovakian local areas due to changes in climate, competition, and human activities, including the clearing and abandonment of land and changes in grazing. In recent decades, the domestic production of black juniper berries was almost completely replaced by imports from Balkan countries.

The presented investigation shows the results of the production capacity of juniper plant populations depending on the size and weight of the fruits. Sufficient attention was carried out to the isolation and content of essential oil and its qualitative–quantitative characteristics. Based on the statistical processing of the data, it can be concluded that there is an α-pinene chemotype of juniper plant populations in Slovakia. From the monitoring of the occurrence of populations during the field research and experimental works, it was confirmed that the genetic diversity at the level of juniper populations in the territory of Slovakia characterises the chemotype species diversity and its structure as an α-pinene chemotype.

In future, great attention must be paid towards the analyses of juniper distillates, which form the basis of the alcoholic beverage, in connection with all already mentioned detections. From the production aspect of juniper berries, the most plentiful is to establish "new" plantations by picking up (digging) young juniper plants from their autochthonous populations and transplanting them to the maintained rural landscape.

**Author Contributions:** I.S. processed the experimental data, performed the biochemical–biological analysis, and drafted the manuscript. P.O. carried out almost all the technical details and performed numerical calculations for the suggested experiment with plant population's collection in Slovakia. M.H. devised the project of Juniper biodiversity, the main conceptual ideas, and proof outline. M.K. and O.K. worked on the hierarchical cluster analysis of results in regard to qualitative–quantitative analyses of juniper essential oil composition. All authors discussed the results and commented on the manuscript. All authors have read and agreed to the published version of the manuscript.

**Funding:** This research was supported by the Slovak Research and Development Agency (SRDA), the project: APVV-14-0843: "R&D of possibilities of growing juniper (*Juniperus communis* L.) for the production of fruits".

**Data Availability Statement:** The data are openly available in a public repository that issues datasets with DOIs.

**Conflicts of Interest:** The authors declare that the research was conducted in the absence of any commercial or financial relationships that could be construed as a potential conflict of interest.

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
