# Peer review of "Selected Biotopes of Juniperus communis L. in Slovakia and Their Chemotype Determination"

_horticulturae, doi:10.3390/horticulturae9060686_

Round 1

Reviewer 1 Report

This study monitored the population of Juniperus communis in Slovakia from 2018 to 2020, and isolated essential oils and identified their qualitative and quantitative characteristics. In general, the investigation was well-performed. It may be published pending some minor revisions.

1) There are a lot of non-English characters in Figures 3-7, which should be in English.

2) The figure legends are missing for Figures 3-7. For example, what do the dotted lines indicate in Figure 4?

3) All figures of quartile ranges should be redrawn with EXCEL or ORIGIN. The rectangles should be in alignment.

4) From Figure 9, I cannot discriminate four different chemotypes.

5) For all tables, statistic analyses (e.g. multiple comparisons) are lacking.

6) Line 174 and Line 312, what does the unit "g.100 pc-1" mean?

There are too many non-English characters.

Author Response

Please, see the attachment_June 02, 2023 ... Word file

Reviewer 2 Report

The manuscript is interesting, useful and relevant for the field. I have made comments in the attached pdf.

The stated “aim of this study is the monitoring of the habitats of common juniper (Juniperus communis L.) in the territory of Slovakia.” No Methods are detailed for this aim. The Results and Discussion sections have no answer for this particular aim, only for the ones that focus on essential oil variability and composition. Either remove the aim or discuss it in the next sections.

The Introduction contains numerous grammatical, spelling and language mistakes that need to be corrected.

Author Response

(The authors gave the same response as above.)

Reviewer 3 Report

The manuscript 'Selected biotopes of Juniper communis L. in Slovakia and their chemotype determination' is well organised and provides interesting insight into certain characteristics of this plant. Please find more detailed comments below.

-  Try to enrich the introduction as per the instructions for authors of the journal.

- Table 3 is not well adjusted in the text.

- Please provide indicators of statistical significance in the tabular data.

- When citing a publication apart from the name(s) of the author(s) the year of the publication should also be mentioned. Please revise accordingly.

- Lines 399-408: I believe it'd be better if you include this paragraph in the discussion section. 

- Reference list includes a few old publications. Try to cite more recent literature.

Minor editing of English language is needed. It'd be of value if you had the manuscript proofread by a native English person.

Author Response

(The authors gave the same response as above.)

Reviewer 4 Report

The manuscript titled 'Selected Biotopes of Juniperus communis L. in Slovakia and their Chemotype Determination' was a very interesting read. The application of compounds derived from natural resources is of crucial importance as renewable alternatives in food, fuel and pharmaceutical industries. This article provides detailed composition of several essential oils and their composition in the Juniperus communis L., highlighting the importance of these findings for the distillery industry.  

I have some comments as follows;

Abstract and Conclusion could be more refined,  abstract ends abruptly - important to continue the flow of the article 

line 28- percentage symbol is missing from the reported findings 

line 50- include reference to the study confirming the diuretic findings stated

line 80 - in period , can be removed to read better

line 106 - may need to include the injection ratio and if it was a split or spitless injection

line 142- repeats what is said in line 138 about the low number of samples (5 each), can it be reworded not to have this info repeated.

Section 3.1 - wording 'facts' should be rewritten as findings.

Figure 3, 4, 5, 6, 7  - graph title, x-axis title etc. should this be in English?

Table 3 - sizing is an issue, missing text

The word "fact" throughout the results section could be replaced with "findings" , reads better

line 231- Can this statement be backed up with a reference 

Paragraph starting at line 256 can be more refined to flow/read better

Suggestion to use abbreviations for the locations ie. Kišovce-Hôrka (KH) and Chrámec-Teplá dolina (CTD) or just simple location 1, location 2 etc. will read better throughout the article

line 315 - expand on how this is confirmed from the findings of this study.

Discussion paragraph starting at line 322 carbohydrates are first mentioned but not in the results section, please distinguish between hydrocarbons and carbohydrates were necessary as these are both very different classes of organic compounds.

line 333 - highlight yields when making this statement

Sentence starting at line 337-339 needs references 

line 360 - should this read as "pathogenic microorganisms' and give examples is possible

Some phrasing throughout the manuscript could be improved 

Author Response

(The authors gave the same response as above.)

Round 2

Reviewer 2 Report

The paper can be published.

Only minor editing needed.